# Management of Cavernous Carotid Artery Aneurysms: A Retrospective Single-Center Experience

**DOI:** 10.3390/brainsci12030330

**Published:** 2022-02-28

**Authors:** Michael Karl Fehrenbach, Eric Dietel, Tim Wende, Johannes Kasper, Caroline Sander, Florian Wilhelmy, Ulf Quaeschling, Juergen Meixensberger, Ulf Nestler

**Affiliations:** 1Department of Neurosurgery, University Hospital Leipzig, 04103 Leipzig, Germany; eric.dietel@medizin.uni-leipzig.de (E.D.); tim.wende@medizin.uni-leipzig.de (T.W.); johannes.kasper@medizin.uni-leipzig.de (J.K.); caroline.sander@medizin.uni-leipzig.de (C.S.); florian.wilhelmy@medizin.uni-leipzig.de (F.W.); juergen.meixensberger@medizin.uni-leipzig.de (J.M.); ulf.nestler@medizin.uni-leipzig.de (U.N.); 2Institute of Neuroradiology, University Hospital Leipzig, 04103 Leipzig, Germany; ulf.quaeschling@medizin.uni-leipzig.de

**Keywords:** cavernous carotid aneurysm, flow diversion, cranial nerve deficits

## Abstract

Objective: While cavernous carotid aneurysms can cause neurological symptoms, their often-uneventful natural course and the increasing options of intravascular aneurysm closure call for educated decision-making. However, evidence-based guidelines are missing. Here, we report 64 patients with cavernous carotid aneurysms, their respective therapeutic strategies, and follow-up. Methods: We included all patients with cavernous carotid aneurysms who presented to our clinic between 2014 and 2020 and recorded comorbidities (elevated blood pressure, diabetes mellitus, and nicotine consumption), PHASES score, aneurysm site, size and shape, therapeutic strategy, neurological deficits, and clinical follow-up. Results: The mean age of the 64 patients (86% female) was 53 years, the mean follow-up time was 3.8 years. A total of 22 patients suffered from cranial nerve deficit. Of these patients, 50% showed a relief of symptoms regardless of the therapy regime. We found no significant correlations between aneurysm size or PHASES score and the occurrence of neurological symptoms. Conclusion: If aneurysm specific symptoms persist over a longer period of time, relief is difficult to achieve despite aneurysm treatment. Patients should be advised by experts in neurovascular centers, weighing the possibility of an uneventful course against the risks of treatment. In this regard, more detailed prospective data is needed to improve individual patient counseling.

## 1. Introduction

Cavernous carotid aneurysms (CCA) account for 2% to 9% of all intracranial aneurysms and are often considered benign due to their low risk of hemorrhage [1]. In addition to the risk of subarachnoid hemorrhage, their location within the sinus cavernous in proximity to cranial nerves can lead to impairment of ocular movement, carotico-cavernous fistula, or epistaxis [2,3,4,5]. Pituitary insufficiency relating to the aneurysm mass effect or evolving after treatment has been described [6]. However, most CCA remain incidental findings, with unspecific symptoms leading to cranial imaging and subsequent detection of the vascular pathology.

There is a high risk that symptoms caused by CCA may only partially resolve [7,8]. A wide variety of neurovascular options exist, ranging from open surgical approaches to endovascular treatment [9,10,11,12]. The recent development of endovascular techniques, especially flow diversion, led to a change of preferred therapy options [13,14]. Although complication rates during endovascular treatment are very low, the life-long antiplatelet therapy poses a specific risk down the line which is to be considered, especially in young patients [15,16,17]. Nevertheless, flow diversion frequently enables definite occlusion.

Only limited data are available to which patient counseling can refer. Van der Schaaf et al. described the outcome of previously published reports on endovascular treatment in their review from 2002 [9]. They already called for observational studies to develop management guidelines. Up to now, no evidence-based guidelines exist which leads to a considerable challenge in neurovascular outpatient clinics. In addition to several case reports and case series, two larger cohorts of patients with CCA have been published. In line with Stiebel-Kalish et al. (185 patients), Kumar et al. published their experience with 39 patients [8,18]. There are also reports on treatment options and the outcome of patients with giant cavernous carotid aneurysms [19,20].

We here report the retrospective data of 64 patients with cavernous carotid aneurysms, focusing on comorbidities, neurological deficits, and their follow-up in the function of aneurysm closure.

## 2. Patients and Methods

A positive vote from the local ethics committee had been obtained before the collection and analysis of patient data (#208-15-01062015). We reviewed our database from 1 January 2014 to 1 November 2020 and identified all patients who presented to the outpatient clinic for a new diagnosis of or during follow-up of CCA. The retrospective overview period covered initial diagnoses between 2001 and 2014. All data were extracted from in-house digitally archived and paper-print patient histories as well as digital image collections. Statistical analysis was performed using the Mann-Whitney-test implemented in Prism 8.4.0 (GraphPad Software, San Diego, CA, USA). Results were considered significant with *p* < 0.05. (* [*p* < 0.05]; ** [*p* < 0.01]; *** [*p* < 0.001]; **** [*p* < 0.0001]).

Comorbidities were recorded in the categories of arterial hypertension, type 2 diabetes mellitus, and nicotine consumption. The PHASES score was calculated for each patient according to Greving et al. [21,22]. Angiographic imaging was reviewed for aneurysm site, size, and shape [23]. The shapes were stratified into three groups: first the “saccular” shape with a narrow neck, second the “irregular” shape with satellite aneurysms or blebs, and third, the fusiform “blister” shape. Clinical symptoms were recorded as given by the repeated neurological examinations during follow-up. Conservative treatment consisted of a wait and see strategy with repeated cranial imaging to monitor changes in aneurysm size. Due to missing treatment guidelines, aneurysm repair was decided on an individual basis for each case considering angiographic data, interdisciplinary consensus, and the patient’s will to be treated.

## 3. Results

### 3.1. Baseline Characteristics

We identified 64 patients (86% female). The mean age of all patients was 53.5 years (Table 1). Arterial hypertension was found in 48.4% of all patients (33.3% in men and 50.9% in women), and 15.6% of patients consumed nicotine (25% of men and 14.5% of women). While 13.3% of female patients suffered from diabetes, no male patients did.

One woman in her 40s suffered from an aneurysmal subarachnoid hemorrhage WFNS Grade 1. We treated her according to national SAH guidelines. Aneurysm closure was achieved by flow diversion. Her clinical course was uneventful, and she was discharged home after 14 days. In 63 patients (98%) the CCA remained unruptured. A total of 23 patients (35.9%) had incidental findings, being asymptomatic for their detected CCA. We detected no difference in patient age or the presence of comorbidities in the comparison of patients with incidental diagnosis to symptomatic patients. However, the PHASES score in the symptomatic group was generally higher than in the asymptomatic group (3.4 ± SD vs. 1.8 ± SD, *p* = 0.075). This was mainly caused by a larger aneurysm diameter (mean 7.4 mm versus 5.1 mm, *p* = 0.12). The one aneurysm bleeding occurred from a saccular aneurysm with a diameter of 23 mm. Of note, one patient did not suffer from any symptoms despite an aneurysm size of 24 mm.

Most aneurysms were of saccular shape with a similar distribution of the different morphologies between both groups. All three fusiform aneurysms became symptomatic with cranial nerve deficits (Table 2).

### 3.2. CND Correlates with the Average Size of CCA

To test the hypothesis, whether cranial nerve deficits are mainly caused by compression of surrounding structures by the CCA, we examined the mean diameter of the aneurysms in patients with and without CND. In the 14 female patients with CND, aneurysm size was 11.8 ± 8.6 mm compared to 5.4 ± 5.5 mm in 52 patients without CND (*p* = 0.118). We thereby cannot demonstrate a significant impact of aneurysm size on cranial nerve deficits.

### 3.3. Treatment of CCA and Outcome

In our series, 6 asymptomatic and 15 symptomatic (headache *n* = 5, CND *n* = 9, dizziness *n* = 1) CCA were treated using coiling, stent-assisted coiling, or flow diversion (Table 2). All patients suffering from headaches, with or without treatment of the aneurysm, showed persistence of this symptom. A total of 50% of all patients suffering from CND showed relief of symptoms. Of note, three out of five (60%) conservatively treated patients demonstrated spontaneous relief of CND, while this applies to only four out of nine (44.4%) of the patients who underwent aneurysm closure (Table 3). Three out of eight patients suffering from dizziness reported complete relief; one of these three underwent aneurysm closure. One out of three (33.3%) patients who reported visual impairment showed spontaneous relief. The mean time for symptom relief was 0.8 months (range: 0.5 to 1 month) in conservatively treated patients and 12 months (range: 5 to 22 months) in patients who underwent closure. The average aneurysm size in patients who experienced symptom relief was smaller than in patients with persistent symptoms (10.3 mm vs. 13.2 mm, *p* = 0.6). This is also true for the PHASES-score, with higher values in patients with persistent deficits (4.4 vs. 7.7, *p* = 0.16). However, neither of these reached statistical significance.

The complication rate of the endovascular treatment was low. Out of 21 patients who underwent treatment, 1 suffered from a carotid-cavernous fistula, which was subsequently treated by coil-embolization. Another patient harboring a very large aneurysm of 24 mm in diameter suffered from an inflammatory response of the vessel wall after coil-embolization, which led to intermittent pituitary insufficiency. No bleeding complication occurred despite platelet aggregation inhibition in 19 patients. In the 43 conservatively treated patients, complications such as additional symptoms, worsening of existing symptoms, or aneurysm bleeding were not observed. The mean follow-up for all patients was 3.8 years, ranging from 6 to 221 months.

## 4. Discussion

Precedent clinical CCA series seldom describe patients with asymptomatic CCA; however, with the increasing availability of CT and MRI, many CCA are discovered incidentally [3,24]. In a recent series, up to 40% of patients are asymptomatic, and the CCA is incidentally found, which matches the observations in our cohort [18,25].

As CT and MRI are non-invasive, they are the diagnostic of choice to evaluate the parasellar region [26,27,28]. CT imaging has the advantage of better visualization of bone erosion while MRI scanning has the potential to delineate a potential intradural extension of the aneurysm since the lateral wall or roof of the cavernous sinus can often be visualized. Angiography remains the gold standard for CCA visualization. It is the most accurate for determining the exact origin, configuration, and flow dynamics of the aneurysm. If surgery or embolization are considered, a precise determination of these points becomes critical.

The incidence of SAH in series of CCA is considered to be very low as most CCA are found in the extra-dural space [29,30,31]. In our cohort, only one patient presented with aneurysm bleeding (1.89%). Other reported vascular events include epistaxis and distal cerebral ischemia due to embolism. We observed neither epistaxis nor distal embolism in our patients. However, patients should still be informed about these rare events and be advised to seek medical help if any of these occur.

Arterial hypertension was found in 48.4% of all patients (33.3% in men and 50.9% in women), which does not differ from the general German population. According to recent population-based studies, the prevalence of elevated blood pressure, including controlled hypertension, in the German population is 47–69% in the age group of 55–64 and 26–60% in the group aged 45–54 [32,33]. In 2013, 34% of German men and 29% of German women over 18 years consumed nicotine on a regular basis [34]. We found comparable rates of smoking in our patients, although only 14.5% of the female patients consumed nicotine.

In our experience, endovascular aneurysm closure was feasible with a low complication rate and without persistent deficit. However, current therapy regimes with flow diversion entail livelong antiplatelet therapy. This is especially relevant for younger patients and needs to be discussed with the respective patients thoroughly; it also necessitates a thorough and strict selection of CCA for occlusion.

Most patients in our study population suffered from headaches, dizziness, and unspecific visual symptoms which initially led to cranial imaging and diagnosis of the aneurysm. It can be assumed that these symptoms were not related to the CCA, as most of these symptoms persisted independent from the therapy regime. On the other hand, patients presenting with CND demonstrated improvement in 50% of cases independent of the therapeutic strategy. Notably, spontaneous symptom relief occurred within a relatively short period of a few weeks. All patients with CND were female. Aneurysm size and therapy regime had no impact on the persistence of symptoms. Taken together, our data neither clarify which patients are at risk of persistent CND nor do they show whether patients with persistent CND benefit from aneurysm closure or not.

Despite shortcomings regarding robust data for these patients, we routinely advise aneurysm closure to patients who suffer from aneurysm-related symptoms. However, the often-present anxiety of the informed patient regarding rare but severe aneurysm complications like bleeding should not be ignored and can prompt a change of the therapy regime, from wait and see to aneurysm closure.

## 5. Conclusions

In this single-center analysis, we confirmed a higher incidence of CCA and consecutive cranial nerve deficits in women. We could not reveal differences between CCA patients and the general population regarding the occurrence of cardiovascular risk factors. Symptom relief after treatment is uncertain, and spontaneous improvement can occur, so patient selection for occlusion and observation is crucial.

## Figures and Tables

**Table 1 brainsci-12-00330-t001:** Patient characteristics.

	All Patients (*n* = 64)	Male (*n* = 9)	Female (*n* = 55) ****
Age			
mean	53.48	49.44 ns	54.15 ns
range	10–84	23–76	10–84
SD	17.76	19.22	17.61
Comorbidities; *n* (%)			
elevated blood pressure	31 (48.4)	3 (33.3) ns	28 (50.9) ns
diabetes mellitus	7 (10.9)	0 ns	7 (12.7) ns
nicotine consumption +	10 (15.6)	2 (25) ns	8 (14.5) *
PHASES-score			
mean	2.8	0.8 ns	3.2 ns
range	0–12	0–4	0–12
Aneurysm diameter (mm)			
mean	6.6	2.7	7.2 *
max. diameter, range	1–29	2–9	1–29
Clinical presentation; *n* (%)			
asymptomatic	23 (35.9)	6 (66.7)	18 (32.7)
headache	16 (25)	2 (22.2)	14 (25.4) □
cranial nerve palsy	14 (21.8)	0	14 (25.4)
dizziness	8 (12.5)	0	8 (14.5)
visual impairment	3 (4.7)	1 (11.1)	2 (3.6)
Treatment; *n* (%)			
conservative	44 (68.7)	9 (100)	35 (63.6)
aneurysm closure	20 (31.3)	0	20 (36.4)

+ 1 Data of only 5 men and 30 women available; □ one patient with headaches experienced subarachnoid hemorrhage. (ns [not significant]; * [*p* < 0.05]; **** [*p* < 0.0001]).

**Table 2 brainsci-12-00330-t002:** Aneurysm characteristics.

	All Aneurysms (*n* = 64)	Asymptomatic Aneurysms (*n* = 23)	Symptomatic Aneurysms (*n* = 41)
Patient age			
mean	53.5	51	54.9
range	10–84	17–77	10–84
SD	17.76	16.9	18.3
Patient comorbidities; *n* (%)			
elevated blood pressure	31 (48.4)	10 (41.7)	21 (52.5)
diabetes mellitus	7 (10.9)	2 (8.3)	5 (12.5)
nicotine consumption	10 (15.6)	4 (16.7)	6 (15)
PHASES			
mean	2.8	1.8 ns	3.4 ns
range	0–12	0–11	0–12
Aneurysm diameter (mm)			
mean	6.6	5.1 ns	7.4 ns
max. diameter, range	1–29	2–21	1–29
max. diameter, mean	7.3	5.5 ns	8.3 ns
1–5 mm	45	17	25
>5–10 mm	6	3	3
>10–20 mm	10	2	8
>20 mm	6	1	5
Aneurysm morphology			
saccular	45	18	27
irregular	7	2	5
blister	9	3	6
fusiform	3	0	3
Treatment; *n* (%)			
conservative	43 (67.2)	17 (73.9)	26 (63.4)
coil	2 (3.1)	1 (4.3)	1 (2.4)
stent	1 (1.6)	1 (4.3)	0
stent and coil	3 (4.7)	0	3 (7.3)
flow diversion	15 (23.4)	4 (17.4)	11 (26.8)
open surgery	0	0	0

(ns [not significant]).

**Table 3 brainsci-12-00330-t003:** Outcome of symptomatic aneurysms.

	Conservative Treatment	Aneurysm Closure
	Relief Yes/No	Relief Yes/No
	*n* = 26	*n* = 15
clinical presentation		
headache	0/11	0/5
cranial nerve palsy	3/2	4/5
dizziness	1/6	0/1
visual impairment	1/2	0/0
time to symptom relief, mean, months	0.8	12

## Data Availability

The raw data of this work is available from the corresponding author (MF) upon reasonable request.

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
