# Peer review of "Management of Cavernous Carotid Artery Aneurysms: A Retrospective Single-Center Experience"

_brainsci, 2022, doi:10.3390/brainsci12030330_

Round 1

Reviewer 1 Report

The authors are reporting their experience concerning carotid cavernous aneurysms (CCA) describing both clinical presentation and management of the lesion.

Although the study does not present groundbreaking novelties, the patients size reported is wide and datas are overall well reported.

Nevertheless I advice few clarifications to be added:

1) authors state that FD treatment is compelled to a lifelong antiplatelet therapy and enables aneurysm occlusion; this statement can be confusing and should be revised. When stent endotelization occurs antiplatelet drugs can be suspended (even in absence of precise guidelines many center relies on 6-12 months of DAPT especially in younger patient). Moreover aneurysm occlusion, although frequently achieved, is not consistent (70-90%).    

2) It is stated that one patient presented with aneurysm haemorrage but this data is not reported in Tab.1; moreover it is not clear what kind of haemorrage the patient experienced (SAH? development of CC fistula?) and how it was managed, please clarify.

3) Since as stated most of the symptoms are caused by aneurysm compression on the cavernous structures I think it would be nice to describe if symptoms relief after treatment appear to be higher in patient where FD is implanted (so that aneurysm sack can shrink) than in patient treated by coiling (in which mass effect is rarely modified).

4) Regarding symptoms it would be important to report pharmacological treatment of these patients, if corticosteroids were employed both in conservative and treatment subgroups and if this affected symptomatology

Author Response

Reviewer #1:

General annotation:

Thank you for the critical reading of our manuscript.

Concern:

Authors state that FD treatment is compelled to a lifelong antiplatelet therapy and enables aneurysm occlusion; this statement can be confusing and should be revised. When stent endotelization occurs antiplatelet drugs can be suspended (even in absence of precise guidelines many center relies on 6-12 months of DAPT especially in younger patient). Moreover, aneurysm occlusion, although frequently achieved, is not consistent (70-90%).

Response:

The patients in our clinic receive a dual antiplatelet therapy using Ticagrelor and Acetylsalicylic acid for at least one year. After a regular one-year-control, ticagrelor medication is ended. The treatment with Acetylsalicylic acid is lifelong. Therefore, we stated „a lifelong antiplatelet therapy “. The phrase „and enables aneurysm occlusion “was altered.

Concern:

It is stated that one patient presented with aneurysm haemorrage but this data is not reported in Tab.1; moreover it is not clear what kind of haemorrage the patient experienced (SAH? development of CC fistula?) and how it was managed, please clarify.

Response:

She presented at our clinic with headaches because of a subarachnoid hemorrhage. She was managed according to national SAH guidelines. Aneurysm closure was achieved by Flow Diversion. Her clinical course was uneventful and she was discharged home after 14 days.

We added this additional information to the results and Tab. 1.

Concern:

Since as stated most of the symptoms are caused by aneurysm compression on the cavernous structures I think it would be nice to describe if symptoms relief after treatment appear to be higher in patient where FD is implanted (so that aneurysm sack can shrink) than in patient treated by coiling (in which mass effect is rarely modified).

Response:

Altogether we treated five patients with Coiling, either Coiling alone or stent-assisted Coiling, only one of them presented with a cranial nerve deficit, which was unchanged after Coiling. Of the 15 patients treated with flow diversion, 5 presented with a cranial nerve deficit. Three of them had a complete or partial relief of the symptoms after flow diversion. So, in accordance with the concern, treatment with flow diversion can bring a relief of symptoms. But for statistical testing a larger cohort of patients treated with Coiling would be needed.

Concern:

Regarding symptoms it would be important to report pharmacological treatment of these patients, if corticosteroids were employed both in conservative and treatment subgroups and if this affected symptomatology.

Response:

None of the patients included in this study received a treatment with corticosteroids.

Reviewer 2 Report

The authors of the manuscript report the retrospective data of 64 patients with cavernous carotid aneurysms, focusing on comorbidities, neurological deficits, and their follow-up in function of aneurysm closure.

My comment (accept after minor revision)

Chapter "2. Patients and Methods" (must be improved)

The authors of the manuscript report that statistical analysis was performed using Prism 8.4.0 GraphPad Software, San Diego, CA, USA. 
Please provide the statistical test used to perform the analysis

Chapter "3. Results" (must be improved)
Next to the name of table 1, please specify what * means - statistics significant when p <??? 

Chapter "5. Conclusions" (can be improved)
I think that in the summary you can add one or two sentences about the management of cavernous carotid artery aneurysms resulting from the observation of the center where the authors work.

Author Response

General Annotations:

Thank you for the critical comments to our work and for carefully revising our manuscript.

Concern:

The authors of the manuscript report that statistical analysis was performed using Prism 8.4.0 GraphPad Software, San Diego, CA, USA. Please provide the statistical test used to perform the analysis.

Response:

The statistical test is now included in the „Patients and Methods “section.

Concern:

Next to the name of table 1, please specify what * means - statistics significant when p <???

Response:

An explanation of the level of significance is now included in the „Patients and Methods “section.

Concern:

I think that in the summary you can add one or two sentences about the management of cavernous carotid artery aneurysms resulting from the observation of the center where the authors work.

Response:

The proposed summary was added.